# Host genetic background and environment have different effects on the establishment and structure of the adult worker honey bee gut microbiota

**Breven Stark**[1], **Gene E. Robinson**[1,2,3☉], **Cassondra L. Vernier**[1,4☉]*

1 Carl R. Woese Institute for Genomic Biology, University of Illinois at Urbana-Champaign, Urbana, Illinois, United States of America, 2 Department of Entomology, University of Illinois at Urbana-Champaign, Urbana, Illinois, United States of America, 3 Neuroscience Program, University of Illinois at Urbana-Champaign, Urbana, Illinois, United States of America, 4 Biology Department, University of Wisconsin-Stout, Menomonie, Wisconsin, United States of America

☉ These authors contributed equally to this work.
* vernierc@uwstout.edu

## Abstract

Many studies have highlighted the importance of gut microbiomes to many aspects of host physiology. Therefore, understanding how factors, such as host genetics and environment, impact the establishment and composition of the gut microbiota can provide insight into host biological functioning. Here, we controlled the microbial inoculation of honey bees across distinct social groups, each composed of a mixture of bees from three distinct genetic backgrounds, and each housed in a distinct cage environment. We determined the relative effects of host genetic background and cage environment on the establishment and composition of the gut microbiota. Using 16S rRNA gene sequencing, we found that host genetic background and cage environment had effects on gut microbiota structure, with each influencing gut microbiota beta diversity. In addition, host genetic background and cage environment each affected the abundance of different individual gut microbes. Together, these results suggest that host genetic background and environment may play distinct roles in shaping the establishment and structure of host gut microbiota.

## Introduction

An increasing body of research highlights the importance of gut microbiomes to host biology. Gut microbiomes have been shown to correlate with host phenotypes in many aspects of animal health and physiology, such as priming host immune response, suppressing pathogen growth in host gastrointestinal tract, influencing host nutrition and digestion, and altering host brain function and behavior [1–7]. Aberrations in the gut microbiota resulting in dysbiosis are also often associated

**Data availability statement:** All sequencing Data is deposited in NIH SRA BioProject (accession number PRJNA1307180). All other data are within the manuscript and its Supporting Information files.

**Funding:** This research was funded by US National Science Foundation Division of Integrative Organismal Systems award number 2120378; PI: GER. The funders did not play any role in the study design, data collection and analysis, decision to publish or preparation of the manuscript.

**Competing interests:** The authors have declared that no competing interests exist.

with adverse health effects for the host, stressing the importance of gut symbionts to health and well-being [1–7]. Therefore, understanding the factors that underlie the establishment and composition of the gut microbiota has important impacts for the understanding of host biology.

Animal microbiomes are typically acquired from contact with conspecifics and the environment. Therefore, many environmental factors, including social interactions, diet, lifestyle, geography and disease, have been associated with variations in gut microbiota between individuals [5,7–10]. In addition, the relationship between gut microbiota and host genetics is well established [11–15]. However, there are challenges to isolating and understanding the effects of host genetics and specific environmental factors on gut microbial community composition due to confounds between each other, the physical environment, and maternal inheritance, which may obscure the full impact of these factors [16,17]. Therefore, studies pinpointing clear roles for host genetics and environment on gut microbial community establishment are limited [14].

The honey bee (*Apis mellifera*) is an excellent model to test the influence of host genetics and environment on gut microbial community establishment and composition, for the following four reasons. First, honey bees are eusocial insects that live in highly structured colonies composed of a single reproductive queen and tens of thousands of non-reproductive worker bees. Colony-like social groups can be maintained in the lab, where even small groups of individuals display similar behavioral interactions, division of labor, and behavioral plasticity as what is seen in the hive [18–21]. This allows for the formation of distinct social groups (environments) under controlled lab conditions, which include groups of interacting individuals and their interactions with the physical environment. Second, in comparison to mammalian models, honey bees have a simple microbiota: the gut is colonized by a stable community of 9 taxonomic clusters, including taxa in Proteobacteria (*Snodgrassella alvi*, *Gilliamella spp.*, *Frischella perrara*, *Bartonella apis*, *Bombella apis,* and *Acetobacteraceae*), Firmicutes (*Bombilactobacillus* Firm-4 and *Lactobacillus* Firm-5), and Actinobacteria (*Bifidobacterium spp.*), which comprise 95%−99.9% of gut bacteria in all individuals [22]. Such relative simplicity allows for ease of quantifying differences in microbiome composition between individuals. Third, honey bees do not experience maternal inheritance of gut microbiota through parturition, as in mammals. Instead, individual adult honey bees acquire their gut microbiome foundation through a combination of chewing through a wax cap covering their brood cell and interacting with older nest mates [23,24]. Therefore, rearing can be manipulated to control for the acquisition, or lack thereof, of the gut microbiota in individual honey bees [25–27]. Fourth, it is possible to manipulate rearing conditions to control for genetic background, due to the haplodiploid sex-determination system and the ability to artificially inseminate honey bee queens [27–29].

There is evidence for effects of both host genetics and social environment on gut microbial community establishment in honey bees [23,24,30,31]. However, because the queen mates with up to 20 different males, typical honey bee colonies consist of a mixture of worker bees from different patrilines [32]. Likewise, typical honey beehives

are composed of a complex physical environment that may itself influence microbiota acquisition [23]. These factors make it difficult to examine the distinct effects of host genetic background and social components of the environment, i.e., social interactions between individuals, on gut microbiota establishment. Here, we parse the effects of host genetics and environment in the establishment of the gut microbiota in worker honey bees using a laboratory-based experimental design that avoids typical challenges when analyzing microbiota inheritance, such as unknown genetic variation or physical environmental factors. In particular, we created mixed genetic-background social groups, each housed in a different cage, and controlled their gut microbiome inoculation [25–27,30]. This allowed us to determine whether gut microbiota establishment differed by genetic background, social group (cage environment), neither, or both.

## Materials and methods

### Honey bee husbandry

We used honey bee colonies that were managed using standard beekeeping techniques at the University of Illinois Bee Research Facility in Urbana, IL. To reduce genetic variation between workers from an individual colony and increase genetic variation between workers from different colonies, we used groups of bees each derived from a queen instrumentally inseminated with sperm from a different single drone (single drone inseminated, or SDI) (queen rearing and inseminations by Megan Mahoney, Mahoney Bees and Queens, Texas/North Dakota). SDI queens produce female worker offspring that are, on average, related by 75% to one another. The bees were a mixture of European subspecies *of Apis mellifera*, primarily *A.m. ligustica*.

### Gut microbiota inoculations

To observe the influence that host genetics and environment have on the establishment and composition of the gut microbiota, we set up controlled rearing cages of a mix of microbiota-depleted honey bees from 3 distinct genetic backgrounds, identified as SDI colonies B4, G1, and W4, and controlled their inoculation with honey bee gut-associated microbes. Under this design, "environments" are represented by a group of interacting bees in a cage, hereafter referred to as "cage environment". While the cage represents a simplified social environment, previous research indicates that small groups of individuals housed in lab cages display behavioral interactions comparable to those observed in natural hive environments [18–21]. To form these social groups, we collected brood frames from the three unrelated SDI colonies established in the summer of 2023. To establish microbiota-depleted bees, we used sterilized forceps to pull tan pupae with dark eyes from their capped cells and then placed them, ventral side up, into sterilized 3D-printed dental-grade resin modular pupation plates [27]. We secured the plates holding the pupae in sterile Plexiglas cages that were stored in an incubator at 34°C and 60–65% relative humidity for 2 days. Once the microbiota-depleted honey bees emerged, we painted each bee with a distinctive color on their thorax that corresponded with their source colony and distributed all bees across 6 sterile Plexiglas cages to create the mixed genetic background groups. We provisioned each cage with a~1 g sterilized pollen patty (pollen source) and 1.7 mL autoclave sterilized inverted microcentrifuge tube containing sterilized 50% sucrose solution. We added bees from each source SDI colony to each cage, such that all cages had 15 bees from colonies B4 and G1, and 9 bees from colony W4.

To control the inoculation of the treatment bees, we fed 4 of the 6 cages with the same gut inoculum for 3 days following previously published methods [27,30]. We prepared inoculum fresh each day using 6 mid- and hind-guts from foragers—identified as individual bees returning to the hive with pollen loads on their hind legs or having a distended abdomen due to nectar loads [33]—from a single typical (non-SDI) honey bee colony. We chilled foragers until immobile, performed gut dissections under sterile conditions, and homogenized 6 mid- and hind-guts using a sterile pestle in 1000 µl PBS. We spun this homogenate at 2800 rpm for 1 minute to separate solid gut material from supernatant, and added 50 µl of the supernatant to 1.7 ml of sterile 25% sucrose solution in 4 of the 6 cages (cages 1–4). As a control, we provisioned 2 of the

6 cages (cages 5–6) with 50 µl of sterile 1xPBS added to sterile 25% sucrose each treatment day. After the third day, we provisioned each cage with sterile 50% sucrose solution for the remainder of the trial. A diagram of social group formation in the cage environment and inoculation is depicted in Fig 1.

After the treatment period, we randomly collected 5 bees of each distinct colony from each cage on day 7. This resulted in 15 bees per cage being sacrificed and analyzed, for a total of 60 treatment (inoculated) bees across 4 cages, and 30 control (non-inoculated) bees across 2 cages. We washed all bees used in 16S rRNA gene sequencing once in 12.5% bleach in water, twice with double deionized water, and then flash froze them. We stored all samples at −80°C until gut dissection, 16S rRNA gene sequencing, and further analysis.

## DNA Extraction and 16S rRNA gene sequencing

We performed DNA extractions and 16S rRNA gene sequencing following previously published methods [27,30]. We dissected frozen honey bee gut samples under the microscope on dry ice using sterile conditions. We homogenized each gut (composed of the mid- and hind-gut) in a PowerSoil Pro Bead Solution tube (Qiagen, Germantown, MD, USA), using a disposable sterile pestle (VWR, Radnor, PA, USA), and extracted DNA using a DNeasy PowerSoil Pro DNA isolation kit (Qiagen), following the manufacturer's instructions. We amplified each sample's hypervariable V4 region of the 16S rRNA gene via PCR amplification in triplicate, with a negative control containing no DNA. We performed PCRs with Platinum Hot Start PCR Master Mix (Invitrogen, Waltham, MA, USA), using primers and barcodes designed in [34] with a final concentration of 0.25 µM, under the following cycling conditions: 94° 3 min, 35x[94° 45s/50° 60s/72° 90s], 72° 10 min. We then used gel electrophoresis to visualize the PCR products on agarose gels to confirm that negative controls did not amplify and that all samples were amplified as expected. No contamination was present, DNA concentrations of negative samples were negligible, and all samples met the prior criteria. We normalized sample concentrations, pooled them, and sequenced them on an Illumina MiSeq with 2x250bp paired-end reads. Sample information is found in S1 Table.

We used QIIME2 to demultiplex the samples, and truncated paired-end reads at the first base with a quality score of <Q3 using DADA2 [35,36]. Using QIIME2, we merged paired-end reads and used DADA2 [36] to identify amplicon sequence variants (ASV). After removing chimeric ASVs, we used a QIIME2 classifier of the V4 region, pretrained using the BEExact database [37], to taxonomically classify the remaining ASVs. Prior to statistical analysis, we removed any ASVs identified as mitochondrial or chloroplast. For ASVs that the BEExact database taxonomically identified as a bee-specific genus but remained unclassified at the species level, we subsequently used NCBI megaBLAST to classify to the species level when possible, as in [27]. To do this, we used the QIIME2 rep-seqs.qza object to obtain each ASV corresponding representative 16S rRNA gene V4 sequences and then manually BLASTed each against the entire NCBI nucleotide database. We assigned unclassified ASVs with the first identified full species with the lowest E-score, under the conditions that query cover > 80% and percent identity > 92%. We labeled any ASV that was only able to be classified to Kingdom Bacteria as "Unclassified." We obtained 1,320,374 (660,187 pairs) sequence reads in total. We then filtered and merged pairs, resulting in 576,690 pairs identified as non-chimeric (87.4%) and 1,012 ASVs identified (S2 Table).

To estimate and analyze the abundance of individual honey bee gut-associated microbial species in each sample, we combined the read counts for all ASVs that matched the same species (for most honey bee gut-associated taxa) or genus (for most non-honey bee gut-associated taxa). For our control (microbiota depleted) bees, we retained all taxonomic groups for analysis, including honey bee gut-associated species and non-honey bee gut-associated genera (S3 Table). However, for visualization, we condensed reads of all non-honey bee gut-associated microbes that made up less than 1% of all samples into a category of "other." For our treated bees, we first analyzed our data retaining all taxonomic groups, however, since many non-honey bee gut-associated genera were not found in a majority of samples (and were thus not significantly different between groups), we condensed reads of most of the non-honey bee gut-associated microbes into a category of "other" and again performed analysis retaining the following honey bee gut-associated taxa: Acetobacteraceae related to *Commensalibacter* [possibly Alpha-2.1 and Alpha-2.2], *Bartonella spp.*, *Bifidobacterium asteroides*,

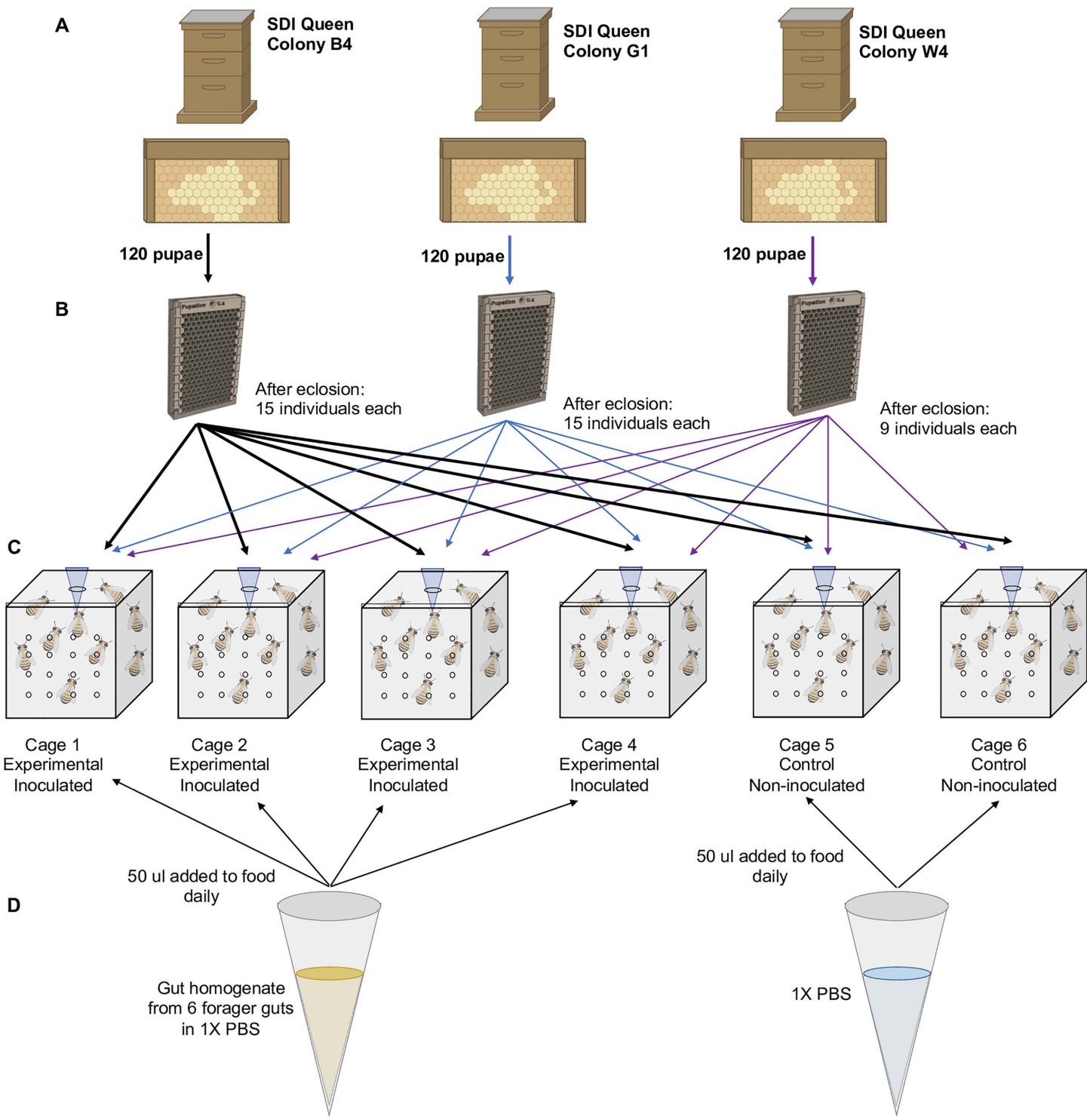

**Fig 1. Diagram of inoculation treatment methods. (A)** Brood frames from 3 colonies headed by unrelated single drone inseminated (SDI) queens were collected and 120 dark-eyed, tan pupae were extracted from each. **(B)** Pupae were placed into sterilized modular pupation plates, which were stored in individual sterilized Plexiglas cages in an incubator until bees eclosed. **(C)** After eclosion, bees were marked with a spot of paint indicative of their source colony and placed into 6 different treatment cages supplied with sterilized sucrose solution and sterilized artificial pollen patties. **(D)** Cages were supplemented with inoculum composed of a gut homogenate (Cages 1-4) or sterile PBS (Cages 5-6) for three days.

*Bifidobacterium coryneforme* [syn. *Bifidobacterium indicum*], *Bombella spp.* [previously *Parasaccharibacter apium*], *Frischella perrara*, *Gilliamella apicola*, *Lactobacillus apis*, *Lactobacillus spp.*, *Lactobacillus helsingborgensis*, *Lactobacillus kullabergensis*, *Lactobacillus melliventris*, *Bombilactobacillus mellifer*, *Bombilactobacillus mellis, Apilactobacillus kunkeei*, and *Snodgrassella alvi* [22,38,39]. We also retained *Fructobacillus spp.* since it was present in all but 1 sample (S1 Table). To calculate the proportion of each taxa in each sample, we divided that taxa's read count by the total read count for each sample. We used raw read counts and/or proportion data in relative abundance analyses as described below.

Since communities may differ in both relative and absolute abundance measures, with each contributing to ecosystem functioning in different ways [40–43], we used both measures of biological diversity [27]. Therefore, in addition to relative abundance, we estimated the absolute abundance of each taxa in each sample ("absolute abundance") by quantifying the bacterial load in each sample using quantitative PCR (qPCR), as in [25,27,44,45]. We performed qPCRs in triplicate as in [27], using primers targeting the 16S rRNA gene or the *A. mellifera Actin* gene as designed in [26], with each primer set including a no template control. Reactions were performed in 384 well plates using a QuantStudio 6 Pro (Applied Biosystems) with the following cycling conditions: 50° 2 min, 95° 10 min, 40x[95° 15 sec/60° 1 min], melt curve: 95° 15 sec, 60° 1 min, 95° 15 sec. We performed standard curves using serial dilutions of plasmids (TOPO pCR2.1 (Invitrogen)) containing the target sequence and measured primer efficiencies using: $E = 10^{(-1/slope)}$ [46]. The copy number of each target (16S rRNA gene or *Actin* gene) in 3 µl of DNA sample was calculated from the sample's $C_t$ score, primer efficiencies, and standard curve using: $n = E^{(intercept-Ct)}$x(DNA extraction elution volume/3) [25,44]. To account for differences in DNA extraction efficiency, we calculated a normalized number of 16S rRNA gene copies by dividing the number of 16S rRNA gene copies by that sample's number of *Actin* gene copies and multiplying by the median number of *Actin* copies [25,44]. We then multiplied the relative abundance (proportion) of each taxa in each sample by the normalized number of 16S rRNA gene copies in that sample to calculate the absolute abundance of each microbial taxa in each sample [25,44]. We then took the $\log_{10}$ value of each of these numbers and used these in further analyses as described below [27].

## Statistical analysis

We performed all statistical analyses in R (v 4.4.0) [47] using previously published methods [27]. All statistics are noted in the figure legends. To analyze gut microbial community composition, we analyzed both alpha diversity (within sample species richness) and beta diversity (between sample species difference) metrics. We analyzed microbiota alpha diversity using Shannon's Diversity Index ("diversity," microbiome package [48]) and a two-way ANOVA ("aov," base statistics package) using colony (genetic background) and cage (environment) as fixed effects, after testing the assumptions of normality using the Shapiro-Wilk Normality Test ("shapiro.test," base statistics package) and homogeneity of variances using Levene's Test ("leveneTest", car package [49]). We visualized this data using boxplots ("geom_boxplot," ggplot2 package [50]).

We analyzed microbiota beta diversity using Permutation MANOVAs with 999 permutations ("adonis2," vegan package [51]) on clr-transformed ("transform," microbiome package [48]) raw read counts as outlined in [52]. We performed post-hoc analysis using Pairwise Permutation MANOVAs with 999 permutations and FDR p-value adjustment ("pairwise.adonis," pairwiseAdonis package [53]), and visualized using Principal Components Analysis ("ordinate," phyloseq package [54]) with Aitchison distance ("distance," phyloseq package [54]), after testing the assumption of uniform dispersion ("betadisper," vegan package [51]).

We analyzed the relative abundance of individual microbes using raw read counts and ANCOM-BC with FDR adjustment ("ancombc2," ANCOMBC package [55,56]) with colony (genetic background) and cage (environment) as a fixed effects, followed by a global test for each effect (using "group" argument in the "ancombc2" function). We analyzed the absolute abundance of individual microbes using Permutation ANOVAs with 999 permutations ("perm.anova," RVAideMemoire package [57]) and we adjusted the p-values multiple comparisons ("p.adjust" with FDR adjustment). We then visualized relative and absolute abundances using proportion data in stacked barplots ("ggplot," ggplot2 package [50]).

## Results

We did not find an effect of genetic background or cage environment on gut microbial alpha diversity in our treatment (inoculated) bees (Fig 2A, all statistics in figure legends). However, we found that both genetic background (Fig 2B) and cage environment (Fig 2C) had significant effects on gut microbiota structure, including influencing beta diversity, as well as the relative and absolute abundance of different individual microbes (Tables 1 and 2, Fig 2D-2H). In particular, genetic background had significant effects on Acetobacteraceae and *F. perrara* relative abundance (Fig 2D-2E, Table 1) and *G. apicola* and *S. alvi* absolute abundance (Fig 2F, Table 2). Cage environment had significant effects on *G. apicola* relative abundance (Fig 2D,2G, Table 1) and *A. kunkeei* relative and absolute abundance (Fig 2D,2G-2H, Tables 1 and 2).

When we analyzed our control (non-inoculated) bees, we found that microbes from the typical 9 taxa clusters that colonize the honey bee gut microbiome were absent or in very low abundance (Fig 3A, Table 3). Instead, these microbiota-depleted bees were colonized by other microbes such as *Bacillus*, *Staphylococcus*, and most frequently, bacteria that were unable to be classified, as is typical of microbiota-depleted honey bees [25,27]. Of particular note is that the microbes present in the control bees were largely absent from the treatment bees (Fig 2, Tables 1 and 2), indicating that they did not establish in the gut in the presence of typical honey bee gut associates.

Even so, when we analyzed the gut microbiota of the control bees, we found that genetic background and cage environment had marginally significant effects on gut microbiota alpha diversity (Fig 3B). As in our treatment bees, control bees differed in gut microbiota beta diversity based upon genetic background (Fig 3C) and cage environment (Fig 3D). When we analyzed the abundance of individual microbes, we found that the abundance of *Neobacillus*, *Paenibacillus*, *Parageobacillus*, *Sporanaerobacter*, *Sporosarcina*, *Tetragenococcus*, and *Turicibacter* differed by genetic background in global statistical tests, but not in pairwise comparisons (Fig 3A, Table 3), while the abundance of *Bifidobacterium asteroides*, *Lactobacillus melliventris*, *Aureimonas*, *Neobacillus*, *Parvimonas,* and *Ureibacillus* differed by cage environment (Fig 3A,3E, Table 3). We did not find that any microbes differed in absolute abundance between our control bees (Table 4).

## Discussion

In this study, we show that host genetics and some social aspects of the environment both shape some components of the host microbiota establishment and composition. Consistent with previous findings [23,30], neither of these factors influenced what taxa established in the gut, measured as alpha diversity (Fig 2A). Rather, they influenced the relative and absolute abundance of a subset of individual microbial taxa (Fig 2B-2E). We found that two honey bee gut-associated microbes (Acetobacteraceae and *F. perrara*) differed in relative abundance, and two honey bee gut-associated microbes (*G. apicola,* and *S. alvi*) differed in absolute abundance between bee genetic backgrounds. We also found that one honey bee gut-associated (*G. apicola*) differed in relative abundance, and one nectar/hive material associated microbe, *A. kunkeei* [22], differed in relative and absolute abundance between cage environments. Some studies explore the function of individual microbes on honey bee biology, including roles of each of these microbes in metabolism [22], *F. perrara* and *S. alvi* in immune responses [58–60], and Acetobacteraceae in protecting the host from fungal pathogens [38,61,62]. However, few studies provide causal evidence [22,25–27,30,31,63,64], and the functions of honey bee gut microbes, especially outside of roles in digestion, nutrition, and immunity, are still largely being explored. Therefore, we currently do not know why host genetics and social group may influence the abundance of some honey bee gut-associated microbes. Nonetheless, given that *A. kunkeei* is not typically considered a honey bee gut symbiont, rather an environmentally acquired microbe [22], it is not surprising that the abundance of this microbe is influenced by the environment.

To understand the effect of host genetic background and social aspects of the environment on the establishment of the gut microbiota, we constructed controlled social groups in sterilized Plexiglas cages (Fig 1). While we did not measure behavioral interactions in the present study, previous studies indicate that small groups of individuals kept in lab cages exhibit similar behavioral interactions as in natural hive settings [18–21]. Likewise, previous studies indicate that although some shifts in gut microbial community occur between caged and hive bees [65], the core gut microbiota is able to be

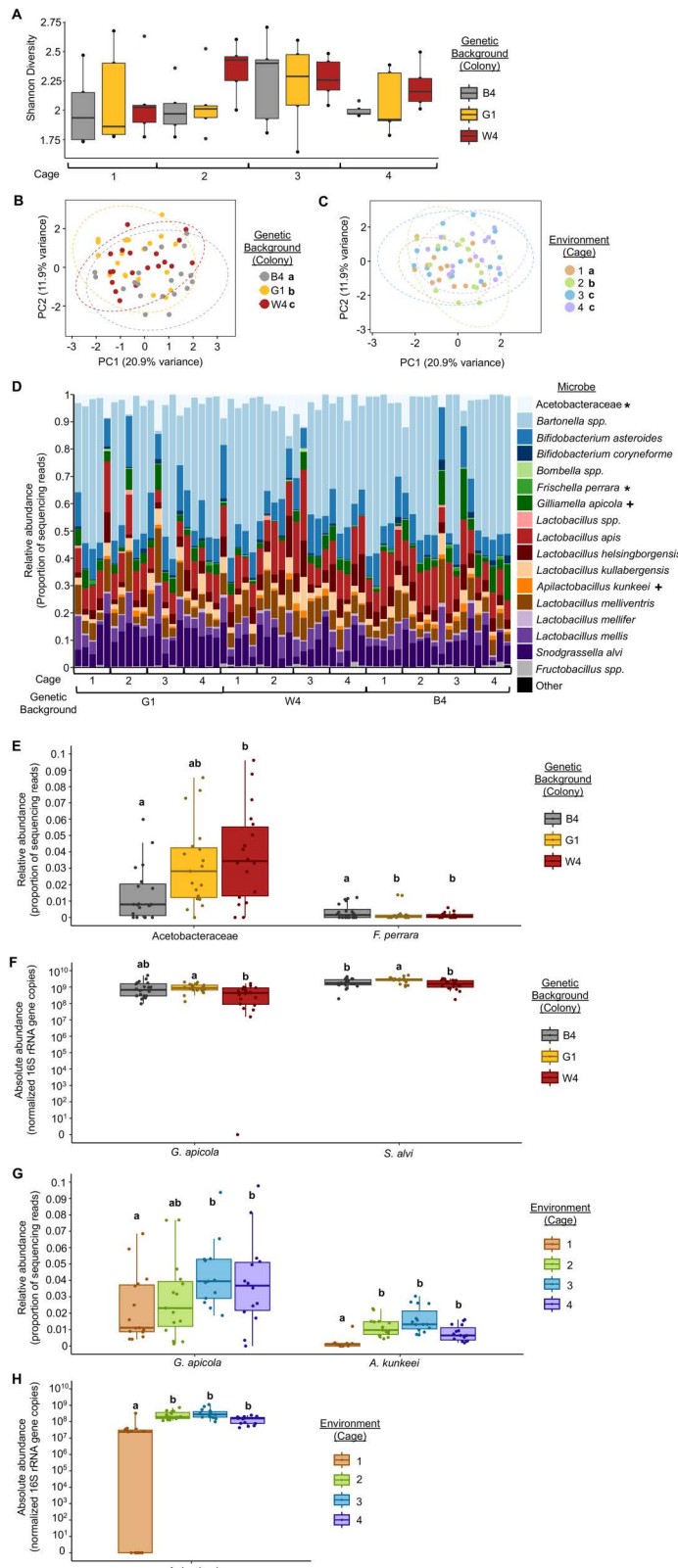

**Fig 2. Host genetic background and social environment influenced gut microbial community structure. (A)** Inoculated bees from different genetic backgrounds and raised in different cage environments did not differ in gut microbiota alpha diversity. Shannon Alpha Diversity: Two-way

ANOVA, Environment: F(3,48) = 1.217, p = 0.314, Genetic Background: F(2,48) = 1.639, p = 0.205, Environment*Genetic Background: F(6, 48) = 0.443, p = 0.846. **(B)** Inoculated bees from different genetic backgrounds differed in gut microbiota beta diversity. Two-way Permutation MANOVA using Aitchison Distance, Genetic Background: F(2,59) = 2.0523, R2 = 0.06392, p = 0.004; Environment: F(3,59) = 2.2910, R2 = 0.10703, p = 0.001; Genetic Background*Environment: F(6,59) = 0.8725, R2 = 0.08153, p = 0.809. n = 20 bees/colony (genetic background), 3 colonies. **(C)** Inoculated bees raised in different cage environments differed in gut microbiota beta diversity. Two-way Permutation MANOVA using Aitchison Distance, Genetic Background: F(2,59) = 2.0523, R2 = 0.06392, p = 0.004; Environment: F(3,59) = 2.2910, R2 = 0.10703, p = 0.001; Genetic Background*Environment: F(6,59) = 0.8725, R2 = 0.08153, p = 0.809. n = 15 bees/cage (social environment), 4 cages. Depicted as Principal Components Analysis (PCA) plots. Lowercase letters in legends denote statistically significant groups via pairwise comparisons. **(D-H)** Inoculated bees from different genetic backgrounds **(D-F)** and raised in different cage environments **(D,G-H)** differed in the relative abundance **(D-E, G)** of four individual microbial taxa and in the absolute abundance **(F, H)** of three individual microbial taxa. **(D)** All taxa depicted as stacked bar plots, with each bar representing a single bee's gut microbial community. Asterisks in legend: *, p < 0.05 ANCOM-BC between genetic backgrounds; +, p < 0.05, ANCOM-BC between social environments. See Table 1 for all p values. **(E-H)** Significant taxa depicted as boxplots, lowercase letters represent statistically significant groups within each microbe via ANCOM-BC (relative abundance) or Permutation ANOVA (absolute abundance). See Tables 1 and 2 for all p values.

**Table 1. Relative abundance of each microbe in the gut microbial communities of treatment bees in each cage environment and genetic background (colony).**

| Microbe | Cage 1 relative abundance (%) | Cage 2 relative abundance (%) | Cage 3 relative abundance (%) | Cage 4 relative abundance (%) | Cage Environment adjusted p value | Colony B4 relative abundance (%) | Colony G1 relative abundance (%) | Colony W4 relative abundance (%) | Genetic Background adjusted p value |
|---|---|---|---|---|---|---|---|---|---|
| Acetobacteraceae | 3.114 | 4.080 | 4.484 | 2.233 | 0.098 | 1.97 | 3.70 | 4.77 | **0.019** |
| *Bartonella spp.* | 42.636 | 37.836 | 30.070 | 44.108 | 0.061 | 42.49 | 38.33 | 35.16 | 0.848 |
| *B. asteroides* | 10.223 | 9.131 | 11.426 | 9.195 | 0.594 | 9.43 | 10.34 | 10.21 | 0.477 |
| *B. coryneforme* | 0.698 | 0.562 | 1.023 | 0.772 | 0.594 | 1.02 | 0.77 | 0.50 | 0.537 |
| *Bombella spp.* | 0.136 | 0.109 | 0.188 | 0.088 | 0.955 | 0.14 | 0.10 | 0.15 | 0.848 |
| *F. perrara* | 0.306 | 0.070 | 0.230 | 0.283 | 0.812 | 0.35 | 0.20 | 0.12 | **0.000** |
| *G. apicola* | 2.315 | 2.886 | 5.849 | 4.472 | **0.001** | 5.13 | 3.99 | 2.53 | 0.185 |
| *Lactobacillus spp.* | 0.153 | 0.265 | 0.056 | 0.000 | 0.955 | 0.11 | 0.17 | 0.08 | 0.804 |
| *L. apis* | 10.554 | 11.950 | 10.464 | 9.221 | 0.812 | 10.97 | 9.33 | 11.34 | 0.753 |
| *L. helsingborgensis* | 5.048 | 5.120 | 4.627 | 5.552 | 0.812 | 5.04 | 3.68 | 6.53 | 0.095 |
| *L. kullabergensis* | 2.893 | 2.848 | 4.135 | 3.188 | 0.899 | 2.64 | 3.18 | 3.98 | 0.186 |
| *A. kunkeei* | 0.149 | 1.143 | 1.595 | 0.777 | **0.000** | 1.03 | 0.76 | 0.97 | 0.401 |
| *L. melliventris* | 6.728 | 7.106 | 8.765 | 5.259 | 0.955 | 5.96 | 7.03 | 7.90 | 0.401 |
| *B. mellifer* | 0.545 | 0.594 | 0.727 | 0.658 | 0.899 | 0.58 | 0.65 | 0.67 | 0.744 |
| *B. mellis* | 5.270 | 4.674 | 5.866 | 3.953 | 0.899 | 3.76 | 5.62 | 5.44 | 0.095 |
| *S. alvi* | 8.626 | 11.174 | 9.812 | 9.537 | 0.594 | 8.69 | 11.73 | 8.94 | 0.744 |
| *Fructobacillus spp.* | 0.542 | 0.415 | 0.617 | 0.623 | 0.899 | 0.58 | 0.39 | 0.67 | 0.401 |

Some microbes differed in relative abundance—the mean percentage of 16S rRNA gene amplicon sequencing reads per sample—between bees in different cage environments and from different genetic backgrounds (colonies). Analyzed via ANCOM-BC with cage and colony as fixed effects, followed by a global test for each effect. p values were adjusted to account for multiple comparisons through FDR adjustment. n = 5 bees/colony/cage, 3 colonies, 4 cages.

recapitulated in lab reared bees [25–27]. Utilizing this experimental design allowed us to control for age, increase genetic diversity between bees originating from different colonies, decrease genetic diversity between bees originating from the same colony, and ensure all bees in each group were only exposed to other bees in the same group, thus forming distinct, albeit similar, social groups and environments. Each cage was administered the same gut inoculum, giving each social group access to the same microbial community, and bees were collected well after the typical period of gut microbiota establishment [22]. Furthermore, the pupal gut microbiota is uncoupled from adult honey bee gut microbiota, indicating

**Table 2.  Absolute abundance of each microbe in the gut microbial communities of treatment bees in each cage environment and genetic background (colony).**

| Microbe | Cage 1 absolute abundance | Cage 2 absolute abundance | Cage 3 absolute abundance | Cage 4 absolute abundance | Cage Environment adjusted *p* value | Colony B4 absolute abundance | Colony G1 absolute abundance | Colony W4 absolute abundance | Genetic Background adjusted *p* value | Cage*Genetic Background adjusted *p* value |
|---|---|---|---|---|---|---|---|---|---|---|
| Acetobacteraceae | 8.177 | 8.699 | 6.645 | 6.893 | 0.545 | 6.417 | 8.389 | 8.004 | 0.275 | 0.973 |
| *Bartonella spp.* | 9.834 | 9.799 | 9.464 | 9.910 | 0.545 | 9.813 | 9.734 | 9.708 | 0.875 | 0.973 |
| *B. asteroides* | 9.341 | 9.300 | 9.345 | 9.223 | 0.673 | 9.272 | 9.361 | 9.273 | 0.561 | 0.973 |
| *B. coryneforme* | 8.106 | 7.542 | 7.196 | 7.527 | 0.966 | 7.763 | 8.205 | 6.810 | 0.198 | 0.973 |
| *Bombella spp.* | 5.490 | 5.454 | 5.547 | 5.342 | 0.977 | 6.724 | 4.103 | 5.549 | 0.198 | 0.966 |
| *F. perrara* | 5.210 | 2.041 | 5.050 | 5.611 | 0.284 | 4.399 | 4.919 | 4.115 | 0.875 | 0.973 |
| *G. apicola* | 8.546 | 8.602 | 9.004 | 8.245 | 0.673 | 8.824 | 8.908 | 8.065 | **0.050** | 0.966 |
| *Lactobacillus spp.* | 1.124 | 1.691 | 0.545 | 0.000 | 0.566 | 0.832 | 1.253 | 0.434 | 0.827 | 0.973 |
| *L. apis* | 9.351 | 9.426 | 9.287 | 9.220 | 0.243 | 9.335 | 9.322 | 9.305 | 0.875 | 0.630 |
| *L. helsingborgensis* | 9.000 | 8.979 | 8.903 | 8.952 | 0.935 | 9.007 | 8.872 | 8.996 | 0.424 | 0.966 |
| *L. kullabergensis* | 8.741 | 8.738 | 8.833 | 8.710 | 0.935 | 8.645 | 8.799 | 8.823 | 0.424 | 0.973 |
| *A. kunkeei* | 4.055 | 8.367 | 8.483 | 8.079 | **0.018** | 7.056 | 7.005 | 7.677 | 0.767 | 0.973 |
| *L. melliventris* | 9.137 | 9.146 | 9.218 | 8.975 | 0.284 | 9.052 | 9.145 | 9.160 | 0.490 | 0.973 |
| *B. mellifer* | 8.059 | 8.085 | 8.135 | 8.051 | 0.935 | 8.041 | 8.142 | 8.065 | 0.683 | 0.966 |
| *B. mellis* | 8.999 | 8.913 | 8.953 | 8.794 | 0.720 | 8.743 | 9.020 | 8.981 | 0.198 | 0.973 |
| *S. alvi* | 9.230 | 9.382 | 9.259 | 9.175 | 0.545 | 9.224 | 9.415 | 9.145 | **0.050** | 0.973 |
| *Fructobacillus spp.* | 8.031 | 7.896 | 8.058 | 7.408 | 0.545 | 8.000 | 7.525 | 8.019 | 0.275 | 0.966 |

Some microbes differed in absolute abundance—the normalized number of 16S rRNA gene copies per sample—between bees in different cage environments and from different genetic backgrounds (colonies). Analyzed via Two-way Permutation ANOVA with cage and colony as fixed effects. *p* values were adjusted to account for multiple comparisons through FDR adjustment. n = 5 bees/colony/cage, 3 colonies, 4 cages.

that prior bacterial influences should play no role in adult gut microbiota establishment [66]. These experimental conditions provided greater control for the initial acquisition of the gut microbial community, allowing our data to exhibit a stronger biological signal. This was likely due to fewer factors contributing to gut microbial structure, such as physical environmental factors.

Our finding that the control group remained microbiota-depleted throughout the study indicates methodological success, namely that the experimental bees bypassed the typical acquisition of microbes during eclosion and did not acquire microbes from any aspect of the physical environment, other than the inoculum. Therefore, any differences in microbiota between treatment bees of different genetic backgrounds and/or cage environments likely result from these factors alone, and likely after acquisition of the microbiota from the inoculum.

Our finding that the microbiota-depleted bees also differed in gut microbiota beta diversity based upon genetic background and cage environment, indicates that these factors may influence the general establishment of microbes in the gut, not just symbionts. However, when we analyzed the abundance of individual microbes, we saw an effect of cage environment on the abundance of several microbes, but did not see an effect of genetic background on any microbes when multiple comparisons were considered. Therefore, it is possible that the effects of genetic background on gut microbial structure are more important when interacting with symbionts, possibly reflecting ancient or ongoing evolutionary interactions, while interactions with non-honey bee gut-associated bacteria are more stochastic and environmentally determined. Indeed, it is likely that the gut microbiota is unstable in microbiota-depleted bees, as indicated by marginally significant differences in gut microbiota alpha diversity, which is typically stable across bees of 7 days of age [22].

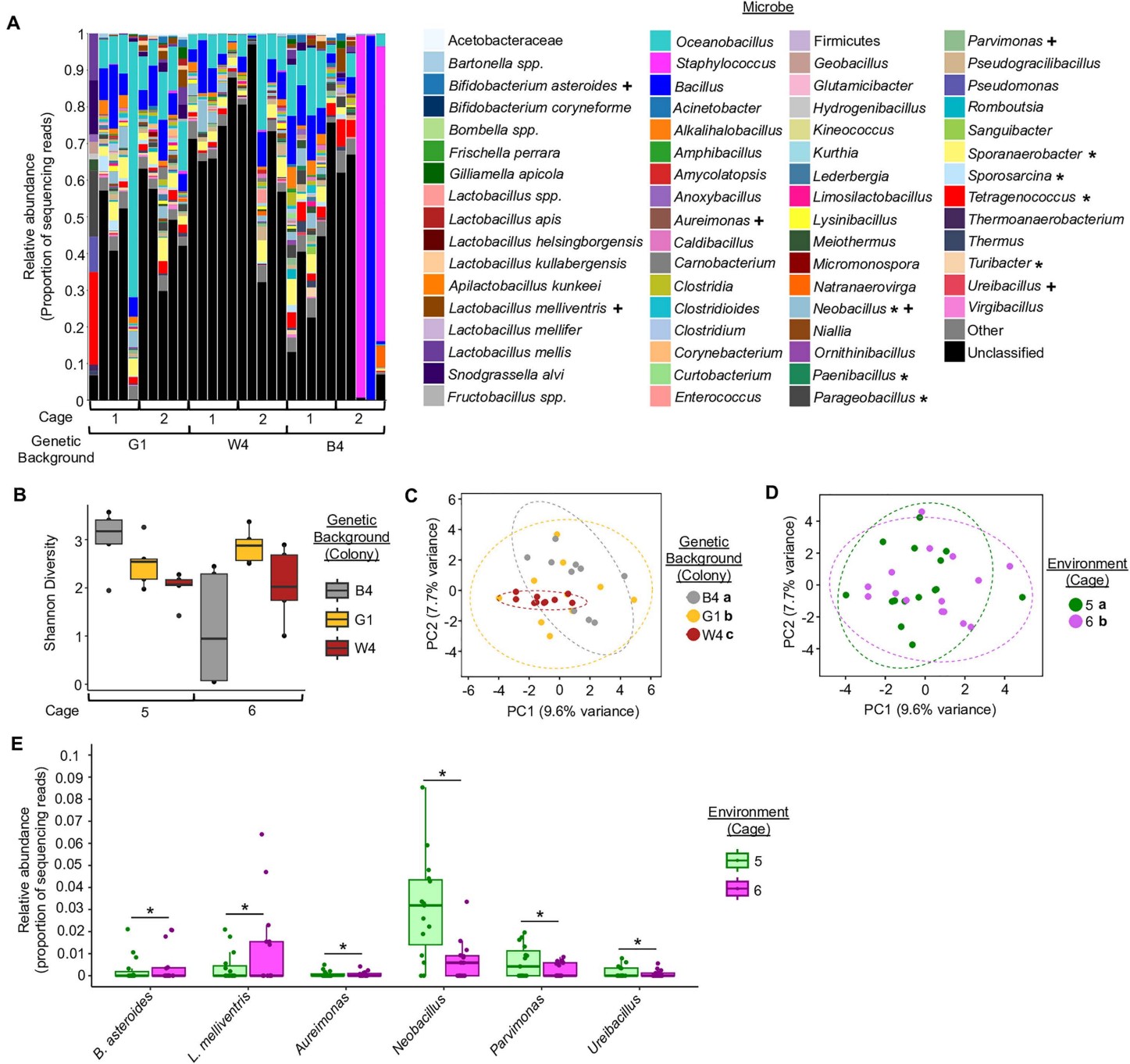

**Fig 3. Control bees remained microbiota-depleted, and showed similar differences in gut microbial community structure as treatment bees.**
**(A)** Control (non-inoculated) bee gut microbiota were mostly composed of non-honey bee gut-associated bacteria, and differed in the abundance of some of these microbes based upon genetic background or cage environment. Depicted as stacked bar plots, with each bar representing a single bee's gut microbial community. "Other" represents all non-honey bee gut-associated microbes that made up less than 1% of all samples. Asterisks in legend: *, $p < 0.05$ ANCOM-BC between genetic backgrounds; +, $p < 0.05$, ANCOM-BC between social environments. See Table 3 for all $p$ values. **(B)** Control bees from different genetic backgrounds and in different cage environments marginally differed in gut microbiota alpha diversity. Shannon Alpha Diversity: Two-way ANOVA, Genetic Background: $F_{(2,24)} = 2.851$, $p = 0.078$, Environment: $F_{(2,24)} = 3.562$, $p = 0.071$, Genetic Background*Environment: $F_{(2,24)} = 7.633$, $p = 0.003$. **(C)** Control bees from different genetic backgrounds differed in gut microbiota beta diversity. Two-way Permutation MANOVA using Aitchison Distance, Genetic Background: $F_{(2,29)} = 1.69$, $R2 = 0.1063$, $p = 0.001$; Environment: $F_{(1,29)} = 1.58$, $R2 = 0.0495$, $p = 0.002$; Genetic

Background*Environment: F(2,29) = 1.428, R2 = 0.0898, p = 0.003. n = 10 bees/colony (genetic background), 3 colonies. **(D)** Control bees raised in different cage environments differed in gut microbiota beta diversity. Two-way Permutation MANOVA using Aitchison Distance, Genetic Background: F(2,29) = 1.69, R2 = 0.1063, p = 0.001; Environment: F(1,29) = 1.58, R2 = 0.0495, p = 0.002; Genetic Background*Environment: F(2,29) = 1.428, R2 = 0.0898, p = 0.003. n = 15 bees/cage (social environment), 2 cages. Depicted as Principal Components Analysis (PCA) plots. Lowercase letters in legends denote statistically significant groups via pairwise comparisons. **(E)** Control bees from different cage environments differed in the relative abundance of six individual microbial taxa. Significant taxa depicted as boxplots, asterisks represent statistically significant differences within each microbe via ANCOM-BC. See Table 3 for all *p* values.

While we only surveyed bees from three distinct genetic backgrounds, across four cage environments, it is possible that other genetic or environmental variations may influence the abundance of microbes not influenced here. In addition, given the larger scale community structure differences between bees from different colonies in other studies [30,67], our findings could also indicate that under natural colony settings, other environmental factors, such as interactions with the physical environment or diet, play a large role in defining gut microbial community structure in bees. This is consistent with previous studies investigating the role of environment on honey bee gut microbiota [23,24]

When interpreting our results in this context, host genetic background appears to be a more robust regulator of some aspects of the gut microbiota than previously proposed. In honey bees, gut microbial communities have been shown to be acquired via social interactions, such as trophallaxis, fecal transfer, or general contact, and vary between individuals in different physical and behavioral environments [22,27,30,44,68,69]. Therefore, it could be expected that social interactions and other environmental factors might homogenize the microbiota between interacting individuals in the same environment. However, we found that, within small experimental social groups, host genetics influenced gut microbial community structure in ways different from the effects of social group (cage environment). Our findings are consistent with those from natural honey bee colony settings, which indicate an association between host genetics and strain-level microbe diversity [31]. Future studies should be able to understand the effects of both genetics and the environment on host health and functioning.

## Supporting information

**S1 Table. Sample Information. Information about all sequencing samples in this study.**
(XLSX)

**S2 Table. ASV counts table. Data containing sequencing counts of each ASV (columns) in each sample (rows).**
(XLSX)

**S3 Table. ASV taxonomy table. Data containing taxonomic classifications of each ASV.**
(XLSX)

**S4 Table. qPCR data. Data containing all Ct scores for the 16S rRNA gene ("16S" tab) and the actin gene ("actin" tab) for each sample, the average Ct scores for each sample ("Ct means" tab), and the normalized number of 16S rRNA gene copies per sample ("absolute" tab). Sequencing Data is deposited in NIH SRA BioProject accession number PRJNA1307180.**
(XLSX)

## Acknowledgments

We thank Amy Cash Ahmed for laboratory assistance, Nathan Beach and Sarah Magdalena Murphree for beekeeping and field assistance, Megan Mahoney for queen rearing and inseminations, Alvaro Hernandez, Chris Wright, and staff at the Carver Biotechnology Center for sequencing services, members of the Computer Network Resource Group of the Carl

**Table 3. Relative abundance of each microbe in the gut microbial communities of control bees in each cage environment and genetic background (colony).**

| Microbe | Cage 5 relative abundance (%) | Cage 6 relative abundance (%) | Cage Environment adjusted p value | Colony B4 relative abundance (%) | Colony G1 relative abundance (%) | Colony W4 relative abundance (%) | Genetic Background adjusted p value |
|---|---|---|---|---|---|---|---|
| Acetobacteraceae | 0.099 | 0.225 | NA | 0.201 | 0.284 | 0.000 | NA |
| Bartonella spp. | 0.297 | 0.297 | 0.493 | 0.286 | 0.334 | 0.271 | 0.802 |
| B. asteroides | 0.291 | 0.442 | **0.008** | 0.606 | 0.494 | 0.000 | NA |
| B. coryneforme | 0.000 | 0.000 | NA | 0.000 | 0.000 | 0.000 | NA |
| Bombella spp. | 0.000 | 0.000 | NA | 0.000 | 0.000 | 0.000 | NA |
| F. perrara | 0.000 | 0.000 | NA | 0.000 | 0.000 | 0.000 | NA |
| G. apicola | 0.123 | 0.407 | NA | 0.343 | 0.448 | 0.004 | NA |
| Lactobacillus spp. | 0.009 | 0.000 | NA | 0.006 | 0.000 | 0.007 | NA |
| L. apis | 0.097 | 0.305 | NA | 0.173 | 0.430 | 0.000 | NA |
| L. helsingborgensis | 0.000 | 0.000 | NA | 0.000 | 0.000 | 0.000 | NA |
| L. kullabergensis | 0.100 | 0.275 | NA | 0.181 | 0.382 | 0.000 | NA |
| A. kunkeei | 0.000 | 0.009 | NA | 0.000 | 0.013 | 0.000 | NA |
| L. melliventris | 0.401 | 1.193 | **0.002** | 1.329 | 1.042 | 0.020 | NA |
| B. mellifer | 0.035 | 0.014 | NA | 0.053 | 0.021 | 0.000 | NA |
| B. mellis | 1.106 | 0.577 | 0.421 | 0.502 | 2.022 | 0.000 | NA |
| S. alvi | 1.361 | 0.629 | 0.560 | 0.641 | 2.338 | 0.004 | NA |
| Enterococcus | 0.016 | 0.291 | NA | 0.000 | 0.069 | 0.392 | NA |
| Fructobacillus | 0.000 | 0.048 | NA | 0.000 | 0.072 | 0.000 | NA |
| Limosilactobacillus | 0.251 | 0.181 | 0.421 | 0.152 | 0.361 | 0.135 | NA |
| Mixta | 0.000 | 0.000 | NA | 0.000 | 0.000 | 0.000 | NA |
| Oceanobacillus | 11.835 | 7.549 | 0.950 | 6.985 | 15.026 | 7.065 | 0.136 |
| Staphylococcus | 0.014 | 12.146 | NA | 17.959 | 0.178 | 0.102 | NA |
| Unclassified | 47.933 | 47.904 | 0.421 | 33.328 | 39.906 | 70.521 | 0.056 |
| Acinetobacter | 0.145 | 0.111 | 0.329 | 0.212 | 0.116 | 0.057 | NA |
| Aeribacillus | 0.037 | 0.032 | NA | 0.058 | 0.047 | 0.000 | NA |
| Aerococcus | 0.000 | 0.048 | NA | 0.023 | 0.000 | 0.048 | NA |
| Agrobacterium | 0.000 | 0.106 | NA | 0.110 | 0.023 | 0.026 | NA |
| Alcaligenes | 0.060 | 0.133 | NA | 0.277 | 0.012 | 0.000 | NA |
| Alkalibaculum | 0.017 | 0.011 | NA | 0.000 | 0.042 | 0.000 | NA |
| Alkalihalobacillus | 2.054 | 0.415 | 0.054 | 1.411 | 1.324 | 0.968 | 0.224 |
| Alkaliphilus | 0.015 | 0.005 | NA | 0.015 | 0.015 | 0.000 | NA |
| Ammoniphilus | 0.083 | 0.000 | NA | 0.000 | 0.125 | 0.000 | NA |
| Amphibacillus | 0.232 | 0.070 | 0.062 | 0.174 | 0.107 | 0.172 | 0.793 |
| Amycolatopsis | 0.067 | 0.173 | NA | 0.050 | 0.115 | 0.196 | NA |
| Anaerocolumna | 0.129 | 0.018 | NA | 0.064 | 0.042 | 0.115 | 0.356 |
| Aneurinibacillus | 0.089 | 0.067 | NA | 0.080 | 0.056 | 0.098 | NA |
| Anoxybacillus | 0.244 | 0.115 | NA | 0.267 | 0.197 | 0.073 | NA |
| Anthophora | 0.077 | 0.048 | 0.054 | 0.063 | 0.124 | 0.000 | NA |
| Atopostipes | 0.022 | 0.010 | NA | 0.015 | 0.000 | 0.034 | NA |
| Aureimonas | 0.078 | 0.073 | **0.014** | 0.024 | 0.156 | 0.044 | NA |
| Bacillaceae | 0.000 | 0.037 | NA | 0.019 | 0.000 | 0.036 | NA |
| Bacillus | 7.418 | 11.436 | 0.847 | 15.146 | 7.550 | 5.585 | 0.377 |
| Blastococcus | 0.063 | 0.015 | NA | 0.074 | 0.043 | 0.000 | NA |
| Brachybacterium | 0.074 | 0.112 | 0.950 | 0.092 | 0.166 | 0.020 | NA |
| Bradyrhizobium | 0.083 | 0.011 | NA | 0.068 | 0.017 | 0.056 | NA |
| Brevibacterium | 0.087 | 0.033 | NA | 0.083 | 0.078 | 0.018 | NA |
| Caldibacillus | 0.318 | 0.158 | 0.421 | 0.279 | 0.175 | 0.260 | 0.356 |
| Caldicellulosiruptor | 0.008 | 0.052 | NA | 0.041 | 0.049 | 0.000 | NA |

*(Continued)*

| Microbe | Cage 5 relative abundance (%) | Cage 6 relative abundance (%) | Cage Environment adjusted p value | Colony B4 relative abundance (%) | Colony G1 relative abundance (%) | Colony W4 relative abundance (%) | Genetic Background adjusted p value |
|---|---|---|---|---|---|---|---|
| *Candidatus* | 0.094 | 0.000 | NA | 0.093 | 0.048 | 0.000 | NA |
| *Carnobacterium* | 0.000 | 0.305 | NA | 0.000 | 0.352 | 0.105 | NA |
| *Catabacter* | 0.076 | 0.096 | 0.591 | 0.000 | 0.101 | 0.158 | NA |
| Christensenellaceae | 0.047 | 0.020 | NA | 0.035 | 0.012 | 0.055 | NA |
| *Clostridia* | 0.712 | 0.709 | 0.267 | 0.898 | 0.689 | 0.544 | 0.356 |
| *Clostridioides* | 0.675 | 0.455 | 0.619 | 0.519 | 0.769 | 0.406 | 0.402 |
| *Clostridium* | 0.353 | 0.114 | 0.304 | 0.207 | 0.350 | 0.144 | 0.356 |
| *Corynebacterium* | 0.130 | 0.251 | 0.054 | 0.058 | 0.316 | 0.199 | NA |
| *Curtobacterium* | 0.092 | 0.034 | NA | 0.138 | 0.000 | 0.051 | NA |
| *Deinococcus* | 0.116 | 0.040 | NA | 0.160 | 0.022 | 0.052 | NA |
| *Desulfohalotomaculum* | 0.079 | 0.000 | NA | 0.077 | 0.007 | 0.035 | NA |
| *Entomomonas* | 0.000 | 0.000 | NA | 0.000 | 0.000 | 0.000 | NA |
| *Escherichia* | 0.077 | 0.085 | NA | 0.109 | 0.033 | 0.100 | NA |
| *Eubacterium* | 0.082 | 0.048 | NA | 0.071 | 0.090 | 0.034 | NA |
| *Firmicutes* | 0.052 | 0.093 | NA | 0.197 | 0.010 | 0.010 | NA |
| *Fundicoccus* | 0.011 | 0.021 | NA | 0.000 | 0.031 | 0.016 | NA |
| *Geobacillus* | 0.457 | 0.076 | NA | 0.370 | 0.407 | 0.022 | NA |
| *Glutamicibacter* | 0.262 | 0.079 | NA | 0.138 | 0.246 | 0.127 | 0.784 |
| *Haematomicrobium* | 0.000 | 0.102 | NA | 0.141 | 0.012 | 0.000 | NA |
| *Haemophilus* | 0.008 | 0.007 | NA | 0.000 | 0.010 | 0.012 | NA |
| *Herbinix* | 0.091 | 0.019 | NA | 0.015 | 0.106 | 0.044 | NA |
| *Hydrogenibacillus* | 0.105 | 0.036 | NA | 0.054 | 0.157 | 0.000 | NA |
| *Irregularibacter* | 0.068 | 0.046 | NA | 0.010 | 0.051 | 0.110 | NA |
| *Janibacter* | 0.019 | 0.037 | NA | 0.000 | 0.016 | 0.067 | NA |
| *Jeotgalicoccus* | 0.027 | 0.010 | NA | 0.021 | 0.000 | 0.035 | NA |
| *Kineococcus* | 0.120 | 0.022 | NA | 0.000 | 0.195 | 0.018 | NA |
| *Klenkia* | 0.009 | 0.010 | NA | 0.000 | 0.015 | 0.013 | NA |
| *Kurthia* | 0.032 | 0.175 | NA | 0.011 | 0.144 | 0.155 | NA |
| *Lachnoclostridium* | 0.089 | 0.097 | NA | 0.023 | 0.197 | 0.059 | NA |
| Lactobacillaceae | 0.000 | 0.000 | NA | 0.000 | 0.000 | 0.000 | NA |
| *Lederbergia* | 0.296 | 0.164 | NA | 0.205 | 0.373 | 0.113 | 0.100 |
| *Leucobacter* | 0.075 | 0.031 | NA | 0.015 | 0.144 | 0.000 | NA |
| *Leuconostoc* | 0.028 | 0.043 | NA | 0.042 | 0.000 | 0.065 | NA |
| *Listeria* | 0.013 | 0.014 | NA | 0.000 | 0.020 | 0.021 | NA |
| *Lysinibacillus* | 0.448 | 0.372 | 0.950 | 0.194 | 0.602 | 0.435 | 0.718 |
| *Marinithermofilum* | 0.032 | 0.082 | NA | 0.000 | 0.037 | 0.133 | NA |
| *Meiothermus* | 0.549 | 0.105 | 0.742 | 0.552 | 0.381 | 0.048 | NA |
| *Methylobacterium* | 0.031 | 0.042 | NA | 0.063 | 0.000 | 0.046 | NA |
| *Methylorubrum* | 0.017 | 0.020 | NA | 0.000 | 0.025 | 0.031 | NA |
| *Micrococcus* | 0.000 | 0.027 | NA | 0.000 | 0.040 | 0.000 | NA |
| *Micromonospora* | 0.000 | 0.148 | NA | 0.000 | 0.142 | 0.079 | NA |
| *Natranaerovirga* | 0.018 | 0.565 | NA | 0.636 | 0.099 | 0.140 | NA |
| *Neobacillus* | 3.066 | 0.661 | **0.002** | 1.411 | 2.091 | 2.088 | **0.050** |
| *Neorhizobium* | 0.008 | 0.021 | NA | 0.000 | 0.043 | 0.000 | NA |
| *Niallia* | 0.092 | 0.351 | NA | 0.051 | 0.369 | 0.243 | NA |
| *Nocardioides* | 0.016 | 0.007 | NA | 0.000 | 0.010 | 0.024 | NA |
| *Nocardiopsis* | 0.074 | 0.065 | 0.476 | 0.063 | 0.088 | 0.057 | 0.492 |
| *Novibacillus* | 0.021 | 0.000 | NA | 0.020 | 0.000 | 0.011 | NA |
| *Ornithinibacillus* | 0.171 | 0.062 | NA | 0.132 | 0.104 | 0.113 | NA |

*(Continued)*

**Table 3.** (Continued)

| Microbe | Cage 5 relative abundance (%) | Cage 6 relative abundance (%) | Cage Environment adjusted p value | Colony B4 relative abundance (%) | Colony G1 relative abundance (%) | Colony W4 relative abundance (%) | Genetic Background adjusted p value |
|---|---|---|---|---|---|---|---|
| *Paenibacillus* | 0.260 | 0.150 | 0.831 | 0.310 | 0.166 | 0.140 | **0.050** |
| *Paeniclostridium* | 0.024 | 0.019 | NA | 0.028 | 0.036 | 0.000 | NA |
| *Paracoccus* | 0.011 | 0.046 | NA | 0.065 | 0.020 | 0.000 | NA |
| *Parageobacillus* | 2.246 | 0.817 | 0.270 | 1.509 | 2.708 | 0.376 | **0.050** |
| *Parasporobacterium* | 0.033 | 0.000 | NA | 0.000 | 0.037 | 0.012 | NA |
| *Parvimonas* | 0.646 | 0.293 | **0.039** | 0.502 | 0.419 | 0.488 | 0.381 |
| *Peribacillus* | 0.032 | 0.000 | NA | 0.036 | 0.012 | 0.000 | NA |
| *Planifilum* | 0.064 | 0.044 | NA | 0.085 | 0.034 | 0.043 | NA |
| *Plantibacter* | 0.052 | 0.021 | NA | 0.047 | 0.062 | 0.000 | NA |
| *Prevotella* | 0.033 | 0.000 | NA | 0.050 | 0.000 | 0.000 | NA |
| *Priestia* | 0.114 | 0.016 | NA | 0.026 | 0.133 | 0.036 | NA |
| *Pseudogracilibacillus* | 0.948 | 0.942 | 0.986 | 0.691 | 0.977 | 1.166 | 0.381 |
| *Pseudokineococcus* | 0.136 | 0.181 | 0.421 | 0.163 | 0.130 | 0.182 | 0.854 |
| *Pseudomonas* | 0.680 | 0.029 | NA | 0.000 | 1.063 | 0.000 | NA |
| *Pseudonocardia* | 0.007 | 0.016 | NA | 0.000 | 0.034 | 0.000 | NA |
| *Psychrobacillus* | 0.049 | 0.025 | NA | 0.026 | 0.037 | 0.048 | NA |
| *Pueribacillus* | 0.000 | 0.075 | NA | 0.000 | 0.092 | 0.020 | NA |
| *Quadrisphaera* | 0.010 | 0.005 | NA | 0.000 | 0.023 | 0.000 | NA |
| *Ralstonia* | 0.103 | 0.083 | 0.847 | 0.151 | 0.020 | 0.108 | NA |
| *Romboutsia* | 0.444 | 0.133 | 1.000 | 0.443 | 0.312 | 0.110 | 0.784 |
| *Rubrobacter* | 0.024 | 0.000 | NA | 0.000 | 0.023 | 0.013 | NA |
| *Rugosimonospora* | 0.028 | 0.015 | NA | 0.000 | 0.022 | 0.042 | NA |
| *Saccharopolyspora* | 0.088 | 0.063 | 0.178 | 0.039 | 0.108 | 0.081 | NA |
| *Salana* | 0.079 | 0.030 | NA | 0.045 | 0.043 | 0.076 | NA |
| *Sanguibacter* | 0.148 | 0.010 | NA | 0.216 | 0.022 | 0.000 | NA |
| *Sedimentibacter* | 0.028 | 0.000 | NA | 0.000 | 0.000 | 0.042 | NA |
| *Snodgrassella* | 0.000 | 0.000 | NA | 0.000 | 0.000 | 0.000 | NA |
| *Solirubrobacter* | 0.024 | 0.032 | NA | 0.000 | 0.000 | 0.085 | NA |
| *Sporanaerobacter* | 3.250 | 2.462 | 0.329 | 2.201 | 3.589 | 2.777 | **0.050** |
| *Sporosarcina* | 0.964 | 0.334 | 0.329 | 0.569 | 1.107 | 0.272 | **0.050** |
| *Streptococcus* | 0.027 | 0.013 | NA | 0.040 | 0.020 | 0.000 | NA |
| *Streptomyces* | 0.000 | 0.025 | NA | 0.006 | 0.000 | 0.032 | NA |
| *Tepidimicrobium* | 0.023 | 0.045 | NA | 0.010 | 0.092 | 0.000 | NA |
| *Tetragenococcus* | 3.486 | 1.610 | 0.311 | 3.025 | 4.006 | 0.613 | **0.050** |
| *Thermoactinomyces* | 0.065 | 0.015 | NA | 0.085 | 0.034 | 0.000 | NA |
| *Thermoanaerobacterium* | 0.133 | 0.013 | NA | 0.021 | 0.186 | 0.011 | NA |
| *Thermus* | 0.547 | 0.379 | 0.476 | 0.758 | 0.454 | 0.177 | 0.370 |
| *Tissierella* | 0.049 | 0.000 | NA | 0.000 | 0.027 | 0.046 | NA |
| *Turicibacter* | 0.409 | 0.174 | 0.742 | 0.405 | 0.391 | 0.078 | **0.050** |
| *Ureibacillus* | 0.168 | 0.091 | **0.039** | 0.115 | 0.159 | 0.113 | NA |
| *Vallitalea* | 0.029 | 0.049 | NA | 0.000 | 0.037 | 0.079 | NA |
| *Virgibacillus* | 0.247 | 0.200 | 0.421 | 0.143 | 0.367 | 0.162 | 0.088 |
| *Weissella* | 0.059 | 0.005 | NA | 0.026 | 0.047 | 0.023 | NA |
| *Xylanivirgra* | 0.029 | 0.000 | NA | 0.043 | 0.000 | 0.000 | NA |

Some microbes differed in relative abundance—the mean percentage of 16S rRNA gene amplicon sequencing reads per sample—between control (non-inoculated) bees in different cage environments and from different genetic backgrounds (colonies). Analyzed via ANCOM-BC with cage and colony as fixed effects, followed by a global test for each effect. *p* values were adjusted to account for multiple comparisons through FDR adjustment. NAs represent taxa that were not present in enough samples to reliably analyze. n = 5 bees/colony/cage, 3 colonies, 2 cages.

**Table 4. Absolute abundance of each microbe in the gut microbial communities of control bees in each cage environment and genetic background (colony).**

| Microbe | Cage 5 absolute abundance | Cage 6 absolute abundance | Cage Environment adjusted *p* value | Colony B4 absolute abundance | Colony G1 absolute abundance | Colony W4 absolute abundance | Genetic Background adjusted *p* value | Cage*Genetic Background adjusted *p* value |
|---|---|---|---|---|---|---|---|---|
| Acetobacteraceae | 0.670 | 2.394 | 0.132 | 1.977 | 2.619 | 0.000 | 0.132 | 0.132 |
| *Bartonella spp.* | 3.026 | 4.113 | 0.592 | 4.208 | 3.621 | 2.879 | 0.773 | 0.874 |
| *B. asteroides* | 1.669 | 1.794 | 0.935 | 2.099 | 3.095 | 0.000 | 0.220 | 0.366 |
| *G. apicola* | 1.254 | 2.082 | 0.592 | 1.577 | 2.636 | 0.792 | 0.618 | 0.392 |
| *Lactobacillus spp.* | 0.542 | 0.000 | 0.783 | 0.385 | 0.000 | 0.429 | 0.980 | 0.874 |
| *L. apis* | 1.048 | 1.432 | 0.792 | 1.058 | 2.663 | 0.000 | 0.220 | 0.874 |
| *L. kullabergensis* | 0.374 | 1.750 | 0.213 | 1.012 | 2.173 | 0.000 | 0.220 | 0.154 |
| *A. kunkeei* | 0.000 | 0.308 | 0.503 | 0.000 | 0.462 | 0.000 | 0.290 | 0.436 |
| *L. melliventris* | 2.090 | 2.289 | 0.935 | 3.265 | 2.867 | 0.435 | 0.220 | 0.220 |
| *B. mellifer* | 0.334 | 0.322 | 1.000 | 0.501 | 0.482 | 0.000 | 0.980 | 0.656 |
| *B. mellis* | 2.146 | 2.114 | 1.000 | 2.484 | 3.906 | 0.000 | 0.132 | 0.811 |
| *S. alvi* | 2.226 | 2.399 | 1.000 | 3.565 | 2.986 | 0.386 | 0.220 | 0.811 |
| *Fructobacillus* | 0.000 | 0.347 | 0.378 | 0.000 | 0.520 | 0.000 | 0.773 | 0.518 |
| *Oceanobacillus* | 5.847 | 5.357 | 0.805 | 4.964 | 5.980 | 5.862 | 0.773 | 0.392 |
| *Staphylococcus* | 0.334 | 2.927 | 0.151 | 2.277 | 1.586 | 1.027 | 0.797 | 0.588 |
| Unclassified | 6.882 | 6.492 | 0.923 | 6.084 | 6.909 | 7.068 | 0.424 | 0.418 |
| *Acinetobacter* | 1.538 | 1.246 | 0.935 | 1.465 | 1.029 | 1.682 | 0.980 | 0.436 |
| *Alkalihalobacillus* | 5.172 | 2.465 | 0.099 | 3.229 | 4.037 | 4.189 | 0.797 | 0.518 |
| *Amphibacillus* | 2.059 | 1.572 | 0.792 | 1.496 | 2.006 | 1.944 | 0.980 | 0.490 |
| *Amycolatopsis* | 1.000 | 1.686 | 0.820 | 0.922 | 1.551 | 1.554 | 0.838 | 0.920 |
| *Anoxybacillus* | 0.745 | 0.678 | 0.935 | 1.030 | 0.576 | 0.529 | 1.000 | 0.777 |
| *Aureimonas* | 0.994 | 1.823 | 0.621 | 0.443 | 2.887 | 0.895 | 0.220 | 0.821 |
| *Bacillus* | 5.797 | 6.066 | 0.935 | 6.141 | 5.798 | 5.856 | 0.980 | 0.811 |
| *Caldibacillus* | 3.026 | 1.375 | 0.378 | 1.587 | 2.071 | 2.943 | 0.773 | 0.582 |
| *Carnobacterium* | 0.000 | 1.106 | 0.403 | 0.000 | 1.113 | 0.546 | 0.717 | 0.740 |
| *Clostridia* | 4.201 | 3.879 | 0.935 | 3.202 | 4.827 | 4.090 | 0.671 | 0.437 |
| *Clostridioides* | 4.260 | 3.479 | 0.592 | 2.705 | 4.886 | 4.017 | 0.424 | 0.392 |
| *Clostridium* | 2.740 | 1.636 | 0.564 | 1.545 | 2.610 | 2.410 | 0.828 | 0.518 |
| *Corynebacterium* | 1.917 | 1.749 | 1.000 | 1.001 | 2.097 | 2.401 | 0.773 | 0.418 |
| *Curtobacterium* | 0.362 | 0.342 | 1.000 | 0.543 | 0.000 | 0.512 | 0.773 | 0.749 |
| *Enterococcus* | 0.322 | 0.771 | 0.783 | 0.000 | 0.556 | 1.083 | 0.797 | 0.874 |
| *Firmicutes* | 0.925 | 0.336 | 0.820 | 1.009 | 0.438 | 0.444 | 0.797 | 0.943 |
| *Geobacillus* | 2.753 | 0.931 | 0.213 | 2.509 | 2.539 | 0.479 | 0.297 | 0.754 |
| *Glutamicibacter* | 2.985 | 0.369 | 0.132 | 1.488 | 2.563 | 0.980 | 0.657 | 0.896 |
| *Hydrogenibacillus* | 0.377 | 0.308 | 1.000 | 0.462 | 0.566 | 0.000 | 0.773 | 0.490 |
| *Kineococcus* | 0.397 | 0.620 | 1.000 | 0.000 | 1.083 | 0.443 | 0.680 | 0.874 |
| *Kurthia* | 0.606 | 1.396 | 0.592 | 0.407 | 1.087 | 1.508 | 0.767 | 0.518 |
| *Lederbergia* | 2.436 | 1.025 | 0.378 | 1.572 | 2.172 | 1.447 | 0.838 | 0.436 |
| *Limosilactobacillus* | 2.683 | 1.713 | 0.621 | 1.026 | 3.683 | 1.884 | 0.290 | 0.811 |
| *Lysinibacillus* | 2.822 | 2.425 | 0.923 | 1.518 | 3.807 | 2.546 | 0.465 | 0.777 |
| *Meiothermus* | 2.091 | 1.307 | 0.592 | 3.009 | 1.577 | 0.513 | 0.220 | 0.198 |
| *Micromonospora* | 0.000 | 1.349 | 0.246 | 0.000 | 1.044 | 0.978 | 0.717 | 0.518 |
| *Natranaerovirga* | 0.299 | 2.049 | 0.292 | 1.536 | 0.982 | 1.004 | 0.917 | 0.985 |

*(Continued)*

**Table 4.** (Continued)

| Microbe | Cage 5 absolute abundance | Cage 6 absolute abundance | Cage Environment adjusted *p* value | Colony B4 absolute abundance | Colony G1 absolute abundance | Colony W4 absolute abundance | Genetic Background adjusted *p* value | Cage*Genetic Background adjusted *p* value |
|---|---|---|---|---|---|---|---|---|
| *Neobacillus* | 5.133 | 2.918 | 0.132 | 2.852 | 5.185 | 4.039 | 0.279 | 0.154 |
| *Niallia* | 1.042 | 2.458 | 0.349 | 0.538 | 2.722 | 1.990 | 0.424 | 0.392 |
| *Ornithinibacillus* | 1.416 | 0.969 | 0.779 | 0.541 | 1.558 | 1.478 | 0.797 | 0.754 |
| *Paenibacillus* | 3.588 | 1.950 | 0.378 | 2.012 | 3.430 | 2.865 | 0.773 | 0.991 |
| *Parageobacillus* | 5.374 | 3.517 | 0.099 | 3.834 | 5.583 | 3.918 | 0.220 | 0.366 |
| *Parvimonas* | 2.542 | 2.444 | 0.974 | 1.665 | 3.260 | 2.554 | 0.767 | 0.392 |
| *Pseudogracilibacillus* | 4.683 | 3.931 | 0.621 | 3.654 | 5.001 | 4.266 | 0.717 | 0.392 |
| *Pseudomonas* | 1.376 | 0.357 | 0.246 | 0.000 | 2.599 | 0.000 | 0.066 | 0.198 |
| *Romboutsia* | 3.399 | 1.376 | 0.132 | 2.544 | 3.635 | 0.985 | 0.220 | 0.198 |
| *Sanguibacter* | 0.703 | 0.314 | 0.923 | 0.600 | 0.924 | 0.000 | 0.828 | 0.811 |
| *Sporanaerobacter* | 5.509 | 4.590 | 0.534 | 4.441 | 5.513 | 5.194 | 0.773 | 0.418 |
| *Sporosarcina* | 3.957 | 2.327 | 0.345 | 2.638 | 3.961 | 2.826 | 0.773 | 0.740 |
| *Tetragenococcus* | 4.962 | 3.671 | 0.386 | 4.607 | 4.787 | 3.557 | 0.773 | 0.198 |
| *Thermoanaerobacterium* | 1.251 | 0.333 | 0.621 | 0.437 | 1.069 | 0.871 | 0.831 | 0.811 |
| *Thermus* | 3.392 | 4.343 | 0.534 | 4.211 | 4.191 | 3.200 | 0.773 | 0.198 |
| *Turicibacter* | 2.112 | 1.981 | 1.000 | 1.583 | 3.222 | 1.334 | 0.671 | 0.811 |
| *Ureibacillus* | 1.371 | 1.617 | 1.000 | 1.007 | 2.477 | 0.999 | 0.714 | 0.436 |
| *Virgibacillus* | 2.709 | 2.012 | 0.741 | 1.921 | 3.691 | 1.468 | 0.429 | 0.728 |

Some microbes differed in absolute abundance—the normalized number of 16S rRNA gene copies per sample—between control (non-inoculated) bees in different cage environments and from different genetic backgrounds (colonies). Analyzed via Two-way Permutation ANOVA with cage and colony as fixed effects. *p* values were adjusted to account for multiple comparisons through FDR adjustment. n = 5 bees/colony/cage, 3 colonies, 2 cages.

R. Woese Institute for Genomic Biology (UIUC) for computational support, Tyler Bauer and the Blugold Center for High Performance Computing (UWEC) for computational support, Zhenqing Chen for the pupation plate design image used in Fig 1, and members of the Robinson and Dolezal laboratories for comments that improved this manuscript.

## Author contributions

**Conceptualization:** Breven Stark, Gene E Robinson, Cassondra L. Vernier.

**Data curation:** Cassondra L. Vernier.

**Formal analysis:** Breven Stark, Cassondra L. Vernier.

**Funding acquisition:** Gene E Robinson.

**Investigation:** Breven Stark, Cassondra L. Vernier.

**Methodology:** Gene E Robinson, Cassondra L. Vernier.

**Project administration:** Gene E Robinson.

**Resources:** Gene E Robinson.

**Supervision:** Gene E Robinson, Cassondra L. Vernier.

**Visualization:** Cassondra L. Vernier.

**Writing – original draft:** Breven Stark, Gene E Robinson, Cassondra L. Vernier.

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
