## [Decision Letter · Decision Letter 0]

28 Sep 2025

Dear Dr. Vernier,

Thank you for submitting your manuscript to PLOS ONE. After careful consideration, we feel that it has merit but does not fully meet PLOS ONE’s publication criteria as it currently stands. Therefore, we invite you to submit a revised version of the manuscript that addresses the points raised during the review process.

We look forward to receiving your revised manuscript.

Kind regards,

Kai Wang

Academic Editor

PLOS ONE

2. Please note that your Data Availability Statement is currently missing [the repository name and/or the DOI/accession number of each dataset OR a direct link to access each database]. If your manuscript is accepted for publication, you will be asked to provide these details on a very short timeline. We therefore suggest that you provide this information now, though we will not hold up the peer review process if you are unable.

4. Please update your submission to use the PLOS LaTeX template. The template and more information on our requirements for LaTeX submissions can be found at http://journals.plos.org/plosone/s/latex

Additional Editor Comments (if provided):

Reviewers' comments:

Reviewer's Responses to Questions

**Comments to the Author**

1. Is the manuscript technically sound, and do the data support the conclusions?

Reviewer #1: Yes

Reviewer #2: Yes

2. Has the statistical analysis been performed appropriately and rigorously?

Reviewer #1: No

Reviewer #2: Yes

3. Have the authors made all data underlying the findings in their manuscript fully available?

Reviewer #1: No

Reviewer #2: No

4. Is the manuscript presented in an intelligible fashion and written in standard English?

Reviewer #1: Yes

Reviewer #2: Yes

Reviewer #1: General comments:

The authors utilized 16S rRNA gene sequencing to investigate how host genetic background and social environment influence the establishment and structure of the gut microbiota in adult worker honey bees. The experimental design is well-controlled, and the manuscript is generally clear and well-organized. The study addresses an interesting and relevant question in host-microbe interactions, and the results contribute to our understanding of microbiota assembly in social insects. I have only a few concerns and suggestions for improvement.

Specific comments/questions:

L 192: I have some reservations about using ANOVA to compare α-diversity and the relative abundance of microbial communities, as the experimental design appears to be a crossover design. Specifically, each hive contains bees from three different genetic backgrounds, and there are four hives in total. This constitutes a typical two-factor experiment, for which a two-way ANOVA or a general linear model might be more appropriate. It is reasonable to expect an interaction effect between hive and genetic background, and we should not artificially ignore this interaction. Of course, my concern is based on the conventions in my own research field, and I am aware that in some disciplines the approach adopted by the authors is standard practice. If that is the case, please proceed in accordance with the accepted methodology in your field.

L 210-214: In my opinion, the placement of this paragraph is suboptimal; it would be more suitably located at the end of the introductory section or at the beginning of the discussion.

L 216-217: The PCoA analysis here would be improved by being complemented with ANOSIM or ADONIS analysis, as this would not only reveal differences based on sample overlap but also demonstrate them from a statistical perspective.

L 312: The term “species” is not appropriate here, as many of your results are at the phylum or genus level; therefore, “taxa” might be a more suitable choice.

Reviewer #2: The scientific work aims to demonstrate how different genetic profiles of Apis mellifera can influence the colonization of specific gut microbiota taxa. The study is very interesting and undoubtedly provides an important contribution to understanding how honey bee genetics can shape the gut microbiota.

Please find below some minor comments and suggestions.

The main concern relates to the use of the concept of “social environment” in this study. To the best of my knowledge and experience, a honey bee cage test cannot truly be considered a social context. In such conditions, the fundamental structure of sociality and colony cohesion is absent: no queen pheromones, no brood pheromones, and no colony tasks or division of labor that normally sustain and shape honey bee society. Nevertheless, “social environment” is presented by the manuscript authors, as a key objective already in the abstract (line 27). I may therefore have misunderstood the intended meaning of this expression, and I kindly ask the authors to clarify how it should be interpreted throughout the manuscript. Please, consider that a cage (line 205), does not constitute a social environment but an (extremely) “simplified environment”.

In general, the abstract could be written more clearly.

The materials and methods part is clear and detailed, but please improve the description of the “social environment” if it is a key point of the study.

It is recommended to include a figure that clearly illustrates the methodology, particularly regarding the SDI queen bees, caging procedures, and microbial inoculations. Such a figure would greatly help readers, especially those who are not accustomed to working with honey bees at this level of detail, to better understand the experimental approach.

Please also clearly state the number of cages (Experimental replicates) per experimental conditions, and the number of bees sacrificed per cage, and the total number of analyzed bees. I think I got this data only in Figure 1D, or from results section (line 253) but it should be clearly stated in the materials and methods section.

Since you did an accurate taxonomical assignment to species level, it is suggested to apply the correction of the 16S rRNA copy number as described by: Raymann, K., Shaffer, Z., & Moran, N. A. (2017). Antibiotic exposure perturbs the gut microbiota and elevates mortality in honeybees. PLoS biology, 15(3), e2001861.

Results section:

The results presented are interesting and partially support the discussion and conclusions of the study. I remain convinced that the conditions tested do not represent a true social environment; rather, they confirm previous findings from other studies on caged bees, which have similarly reported imbalances in the microbiome under such artificial conditions. It is noteworthy that significant effects are shown for some microbial taxa when considering the genetic backgrounds, such as Frischella and Acetobacteriaceae. This is an important aspect that deserves further discussion, and I would encourage the authors to expand on it.

In addition, in Table 2, please clarify more explicitly that cages 5 and 6 represent the non-inoculated cages, and therefore serve as the experimental controls.

Discussion:

Lines 355-356: the statement and assumption is too strong. And there is plenty of evidence of tests performed in a cage, which unfortunately did not provide the same output in the field. This is why, when possible, the semi-field tests are envisaged and preferred to cage tests.

In general, I noticed that the discussion does not fully take into account several relevant and impactful studies on the honey bee microbiota. To strengthen the interpretation of your findings, I would kindly suggest broadening the range of references considered. Important and insightful contributions have been published by researchers worldwide, not only in highly ranked journals, and integrating these perspectives could provide additional depth and context to your discussion.

**Do you want your identity to be public for this peer review?** For information about this choice, including consent withdrawal, please see our Privacy Policy

Reviewer #1: No

Reviewer #2: No

---

## [Author Response · Author response to Decision Letter 1]

3 Dec 2025

Reviewer #1: General comments:

The authors utilized 16S rRNA gene sequencing to investigate how host genetic background and social environment influence the establishment and structure of the gut microbiota in adult worker honey bees. The experimental design is well-controlled, and the manuscript is generally clear and well-organized. The study addresses an interesting and relevant question in host-microbe interactions, and the results contribute to our understanding of microbiota assembly in social insects. I have only a few concerns and suggestions for improvement.

Specific comments/questions:

L 192: I have some reservations about using ANOVA to compare α-diversity and the relative abundance of microbial communities, as the experimental design appears to be a crossover design. Specifically, each hive contains bees from three different genetic backgrounds, and there are four hives in total. This constitutes a typical two-factor experiment, for which a two-way ANOVA or a general linear model might be more appropriate. It is reasonable to expect an interaction effect between hive and genetic background, and we should not artificially ignore this interaction. Of course, my concern is based on the conventions in my own research field, and I am aware that in some disciplines the approach adopted by the authors is standard practice. If that is the case, please proceed in accordance with the accepted methodology in your field.

We thank the reviewer for this comment, and fully agree. Our manuscript uses a two-way ANOVA to compare alpha diversity and ANCOM-BC to analyze the relative abundance of individual microbes. These tests and statistics were included in the methods & figure legend. We have added a statement in the methods and results to reflect this.

Line 230: “All statistics are noted in the figure legends.”

Lines 255-256: “We did not find an effect of genetic background or cage environment on gut microbial alpha diversity in our treatment (inoculated) bees (Fig 2A, all statistics in figure legends).”

We also noticed that we included a typo in the methods section. We have fixed this typo to reflect our use of a two-way ANOVA for these data.

Line 232-234: “We analyzed microbiota alpha diversity using Shannon’s Diversity Index (“diversity,” microbiome package [49]) and a two-way ANOVA (“aov,” base statistics package) using colony (genetic background) and cage (environment) as a fixed effects…”

L 210-214: In my opinion, the placement of this paragraph is suboptimal; it would be more suitably located at the end of the introductory section or at the beginning of the discussion.

This is a great suggestion, and we have moved this paragraph to the end of the introduction.

Lines 86-89: “In particular, we created mixed genetic-background social groups, each housed in a different cage, and controlled their gut microbiome inoculation [25–27,30]. This allowed us to determine whether gut microbiota establishment differed by genetic background, social group (cage environment), neither, or both.”

L 216-217: The PCoA analysis here would be improved by being complemented with ANOSIM or ADONIS analysis, as this would not only reveal differences based on sample overlap but also demonstrate them from a statistical perspective.

We thank the reviewer for this comment. We included a permutation ANOVA (ADONIS) for these data. The results are included in the Figure 2 legend:

Lines 270-273: “Two-way Permutation MANOVA using Aitchison Distance, Genetic Background: F(2,59) = 2.0523, R2 = 0.06392, p = 0.004; Environment: F(3,59) = 2.2910, R2 = 0.10703, p = 0.001; Genetic Background*Environment: F(6,59) = 0.8725, R2 = 0.08153, p = 0.809. n = 20 bees/colony (genetic background), 3 colonies.”

L 312: The term “species” is not appropriate here, as many of your results are at the phylum or genus level; therefore, “taxa” might be a more suitable choice.

Thank you for this comment. We agree and have changed this term to “taxa.”

Lines 345-346: “Control bees from different cage environments differed in the relative abundance of six individual microbial taxa.”

Reviewer #2: The scientific work aims to demonstrate how different genetic profiles of Apis mellifera can influence the colonization of specific gut microbiota taxa. The study is very interesting and undoubtedly provides an important contribution to understanding how honey bee genetics can shape the gut microbiota.

Please find below some minor comments and suggestions.

The main concern relates to the use of the concept of “social environment” in this study. To the best of my knowledge and experience, a honey bee cage test cannot truly be considered a social context. In such conditions, the fundamental structure of sociality and colony cohesion is absent: no queen pheromones, no brood pheromones, and no colony tasks or division of labor that normally sustain and shape honey bee society. Nevertheless, “social environment” is presented by the manuscript authors, as a key objective already in the abstract (line 27). I may therefore have misunderstood the intended meaning of this expression, and I kindly ask the authors to clarify how it should be interpreted throughout the manuscript. Please, consider that a cage (line 205), does not constitute a social environment but an (extremely) “simplified environment”.

We thank the reviewer for this comment, and agree that this is a simplified environment. We have therefore removed use of “social environment” from the text and have replaced with “environment” or “cage environment” throughput the text (denoted in tracked document). We have also changed the title accordingly.

We have also added a statement to the methods section to describe our use of the term “environment” as below.

Lines 106-110: “Under this design, “environments” are represented by a group of interacting bees in a cage, hereafter referred to as “cage environment”. While the cage represents a simplified social environment, previous research indicates that small groups of individuals housed in lab cages display behavioral interactions comparable to those observed in natural hive environments [18–21].”

In general, the abstract could be written more clearly.

The materials and methods part is clear and detailed, but please improve the description of the “social environment” if it is a key point of the study.

We thank the reviewer for this comment and have changed the abstract to address this as follows:

Lines 25-29: “Here, we controlled the microbial inoculation of honey bees across distinct social groups, each composed of a mixture of bees from three distinct genetic backgrounds, and each housed in a distinct cage environment. We determined the relative effects of host genetic background and cage environment on the establishment and composition of the gut microbiota. Using 16S rRNA gene sequencing, we found that host genetic background and cage environment had effects on gut microbiota structure, with each influencing gut microbiota beta diversity. In addition, host genetic background and cage environment each affected the abundance of different individual gut microbes. Together, these results suggest that host genetic background and environment may play distinct roles in shaping the establishment and structure of host gut microbiota.”

It is recommended to include a figure that clearly illustrates the methodology, particularly regarding the SDI queen bees, caging procedures, and microbial inoculations. Such a figure would greatly help readers, especially those who are not accustomed to working with honey bees at this level of detail, to better understand the experimental approach.

We thank the reviewer for this suggestion. We have included this figure and direct readers to it in our methods section:

Lines 134-135: “A diagram of social group formation in the cage environment and inoculation is depicted in Fig 1.”

Lines 137-144: “Fig 1. Diagram of inoculation treatment methods. (A) Brood frames from 3 colonies headed by unrelated single drone inseminated (SDI) queens were collected and 120 dark-eyed, tan pupae were extracted from each. (B) Pupae were placed into sterilized modular pupation plates, which were stored in individual sterilized Plexiglas cages in an incubator until bees eclosed. (C) After eclosion, bees were marked with a spot of paint indicative of their source colony and placed into 6 different treatment cages supplied with sterilized sucrose solution and sterilized artificial pollen patties. (D) Cages were supplemented with inoculum composed of a gut homogenate (Cages 1-4) or sterile PBS (Cages 5-6) for three days.”

Please also clearly state the number of cages (Experimental replicates) per experimental conditions, and the number of bees sacrificed per cage, and the total number of analyzed bees. I think I got this data only in Figure 1D, or from results section (line 253) but it should be clearly stated in the materials and methods section.

Thank you for this comment. We have added the following statements to the methods section:

Lines 123-124: “we fed 4 of the 6 cages with the same gut inoculum for 3 days…”

Lines 145-147: “After the treatment period, we randomly collected 5 bees of each distinct colony from each cage on day 7. This resulted in 15 bees per cage being sacrificed and analyzed, for a total of 60 treatment (inoculated) bees across 4 cages, and 30 control (non-inoculated) bees across 2 cages.”

Since you did an accurate taxonomical assignment to species level, it is suggested to apply the correction of the 16S rRNA copy number as described by: Raymann, K., Shaffer, Z., & Moran, N. A. (2017). Antibiotic exposure perturbs the gut microbiota and elevates mortality in honeybees. PLoS biology, 15(3), e2001861.

We thank the reviewer for this comment and have added this analysis. We include these data in Figure 2, Tables 1-2, Table S4, and outline these methods and results as below:

Lines 206-226: “Since communities may differ in both relative and absolute abundance measures, with each contributing to ecosystem functioning in different ways [40–43], we used both measures of biological diversity [27]. Therefore, in addition to relative abundance, we estimated the absolute abundance of each taxa in each sample (“absolute abundance”) by quantifying the bacterial load in each sample using quantitative PCR (qPCR), as in [25,27,44,45]. We performed qPCRs in triplicate as in [27] using primers targeting the 16S rRNA gene or the A. mellifera Actin gene as designed in [46], with each primer set including a no template control. Reactions were performed in 384 well plates using a QuantStudio 6 Pro (Applied Biosystems) with the following cycling conditions: 50° 2 min, 95° 10 min, 40x[95° 15 sec/60° 1 min], melt curve: 95° 15 sec, 60° 1 min, 95° 15 sec. We performed standard curves using serial dilutions of plasmids (TOPO pCR2.1 (Invitrogen)) containing the target sequence and measured primer efficiencies using: E = 10(-1/slope) [47]. The copy number of each target (16S rRNA gene or Actin gene) in 3 �l of DNA sample was calculated from the sample’s Ct score, primer efficiencies, and standard curve using: n = E(intercept-Ct)x(DNA extraction elution volume/3) [25,44]. To account for differences in DNA extraction efficiency, we calculated a normalized number of 16S rRNA gene copies by dividing the number of 16S rRNA gene copies by that sample’s number of Actin gene copies and multiplying by the median number of Actin copies [25,44]. We then multiplied the relative abundance (proportion) of each taxa in each sample by the normalized number of 16S rRNA gene copies in that sample to calculate the absolute abundance of each microbial taxa in each sample [25,44]. We then took the log10 value of each of these numbers and used these in further analyses as described below [27].”

Lines 248-252: “We analyzed the absolute abundance of individual microbes using Permutation ANOVAs with 999 permutations (“perm.anova,” RVAideMemoire package [58]) and we adjusted the p-values multiple comparisons (“p.adjust” with FDR adjustment). We then visualized relative and absolute abundances using proportion data in stacked barplots (“ggplot,” ggplot2 package [51]).”

Lines 257-263: “However, we found that both genetic background (Fig 2B) and cage environment (Fig 2C) had significant effects on gut microbiota structure, including influencing beta diversity, as well as the relative and absolute abundance of different individual microbes (Tables 1-2, Fig 2D-H). In particular, genetic background had significant effects on Acetobacteraceae and F. perrara relative abundance (Fig 2D-E, Table 1) and G. apicola and S. alvi absolute abundance (Fig 2F, Table 2). Cage environment had significant effects onG. apicola relative abundance (Fig 2D,G, Table 1) and A. kunkeei relative and absolute abundance (Fig 2D,G-H, Table 1, Table 2).”

Lines 280-282 (Fig 2 legend): “(D-H) Inoculated bees from different genetic backgrounds (D-F) and raised in different cage environments (D,G-H) differed in the relative abundance (D-E, G) of four individual microbial taxa and in the absolute abundance (F, H) of three individual microbial taxa.”

Lines 298-304: “Table 2. Absolute abundance of each microbe in the gut microbial communities of treatment bees in each cage environment and genetic background (colony). Some microbes differed in absolute abundance—the normalized number of 16S rRNA gene copies per sample—between bees in different cage environments and from different genetic backgrounds (colonies). Analyzed via Two-way Permutation ANOVA with cage and colony as fixed effects. p values were adjusted to account for multiple comparisons through FDR adjustment. n = 5 bees/colony/cage, 3 colonies, 4 cages.”

Lines 321-322: “We did not find that any microbes differed in absolute abundance between our control bees (Table 4).”

Lines 359-365: “Table 4. Absolute abundance of each microbe in the gut microbial communities of control bees in each cage environment and genetic background (colony). Some microbes differed in absolute abundance—the normalized number of 16S rRNA gene copies per sample—between control (non-inoculated) bees in different cage environments and from different genetic backgrounds (colonies). Analyzed via Two-way Permutation ANOVA with cage and colony as fixed effects. p values were adjusted to account for multiple comparisons through FDR adjustment. n = 5 bees/colony/cage, 3 colonies, 2 cages.”

Lines 371-377: “Rather, they influenced the relative and absolute abundance of a subset of individual microbial taxa (Fig 2B-E). We found that two honey bee gut-associated microbes (Acetobacteraceae and F. perrara) differed in relative abundance, and two honey bee gut-associated microbes (G. apicola, and S. alvi) differed in absolute abundance between bee genetic backgrounds. We also found that one honey bee gut-associated (G. apicola) differed in relative abundance, and one nectar/hive material associated microbe, A. kunkeei [22], differed in relative and absolute abundance between cage environments.”

Lines 647-649: “S4 Table. qPCR data. Data containing all Ct scores for the 16S rRNA gene (“16S” tab) and the actin gene (“actin” tab) for each sample, the average Ct scores for each sample (“Ct means” tab), and the normalized number of 16S rRNA gene copies per sample (“absolute” tab).”

Results section:

The results presented are interesting and partially support the discussion and conclusions of the study. I remain convinced that the conditions tested do not represent a true social environment; rather, they confirm previous findings from other studies on caged bees, which have similarly reported imbalances in the microbiome under such artificial conditions. It is noteworthy that significant effects are shown for some microbial taxa when considering the genetic backgrounds, such as Frischella and Acetobacteriaceae. This is an important aspect that deserves further discussion, and I would encourage the authors to expand on it.

We have added the following statements to the discussion in response to this comment:

Lines 368-369: “In this study, we show that host genetics and some social aspects of the environment both shape some components of the host microbiota establishment and composition”

Lines 377-382: “Some studies explore the function o

---

## [Decision Letter · Decision Letter 1]

18 Dec 2025

Host genetic background and environment have different effects on the establishment and structure of the adult worker honey bee gut microbiota

PONE-D-25-50363R1

Dear Dr. Vernier,

We’re pleased to inform you that your manuscript has been judged scientifically suitable for publication and will be formally accepted for publication once it meets all outstanding technical requirements.

Kind regards,

Kai Wang

Academic Editor

PLOS One

Additional Editor Comments (optional):

Reviewers' comments:

Reviewer's Responses to Questions

**Comments to the Author**

Reviewer #1: All comments have been addressed

Reviewer #2: All comments have been addressed

2. Is the manuscript technically sound, and do the data support the conclusions?

Reviewer #1: Yes

Reviewer #2: Yes

3. Has the statistical analysis been performed appropriately and rigorously?

Reviewer #1: Yes

Reviewer #2: Yes

4. Have the authors made all data underlying the findings in their manuscript fully available?

Reviewer #1: Yes

Reviewer #2: Yes

5. Is the manuscript presented in an intelligible fashion and written in standard English?

Reviewer #1: Yes

Reviewer #2: (No Response)

Reviewer #1: I am satisfied with the authors' revisions and responses to my comments, and I believe the paper now meets the criteria for publication.

Reviewer #2: (No Response)

**Do you want your identity to be public for this peer review?** For information about this choice, including consent withdrawal, please see our Privacy Policy

Reviewer #1: No

Reviewer #2: No

---

## [Editor Report · Acceptance letter]

PONE-D-25-50363R1

PLOS One

Dear Dr. Vernier,

I'm pleased to inform you that your manuscript has been deemed suitable for publication in PLOS One. Congratulations! Your manuscript is now being handed over to our production team.

Kind regards,

on behalf of

Dr. Kai Wang

Academic Editor

PLOS One